# Ratio of extracellular water to intracellular water and simplified creatinine index as predictors of all-cause mortality for patients receiving hemodialysis

**Takahiro Yajima** [1]*, **Kumiko Yajima**[2]

**1** Department of Nephrology, Matsunami General Hospital, Gifu, Japan, **2** Department of Internal Medicine, Matsunami General Hospital, Gifu, Japan

* yajima5639@gmail.com

**Data Availability Statement:** All relevant data are within the paper and its Supporting information files.

## Abstract

The bioelectrical impedance analysis-measured ratio of extracellular water (ECW) to intracellular water (ICW) reflects fluid volume and malnutrition. It may be an indicator of protein-energy wasting and muscle wasting in hemodialysis patients. We examined the association between the ECW/ICW ratio and simplified creatinine index, which is a new surrogate marker of protein-energy wasting and muscle wasting, and whether their combination can accurately predict mortality. A total of 224 patients undergoing hemodialysis for more than 6 months and having undergone bioelectrical impedance analysis for the assessment of body composition were included. Patients were divided into two groups based on the cut-off values of the ECW/ICW ratio (0.57) and simplified creatinine index (20.4 mg/kg/day) for maximumly predicting mortality. Thereafter, they were cross-classified into four groups with each cut-off point. The ECW/ICW ratio was independently associated with the simplified creatinine index (β = -0.164; P = 0.042). During a follow-up of 3.5 years (2.0–6.0 years), 77 patients died. A higher ECW/ICW ratio (adjusted hazard ratio, 3.66, 95% confidence interval 1.99–6.72, P <0.0001) and lower simplified creatinine index (adjusted hazard ratio, 2.25, 95% confidence interval 1.34–3.79, P = 0.0021) were independently associated with an increased risk of all-cause mortality. The adjusted hazard ratio for the higher ECW/ICW ratio and lower simplified creatinine index group vs. the lower ECW/ICW ratio and higher simplified creatinine index group was 12.22 (95% confidence interval 3.68–40.57, p <0.0001). Furthermore, the addition of the ECW/ICW ratio and simplified creatinine index to the baseline risk model significantly improved the C-index from 0.831 to 0.864 (p = 0.045). In conclusion, the ECW/ICW ratio may be a surrogate marker of muscle wasting. Moreover, combining the ECW/ICW ratio and simplified creatinine index may improve the accuracy of predicting all-cause mortality and help stratify the mortality risk of hemodialysis patients.

**Funding:** The authors received no specific funding for this work. The funders had no role in study design, data collection and analysis, decision to publish, or preparation of the manuscript.

**Competing interests:** The authors have declared that no competing interests exist.

## Introduction

Fluid overload and malnutrition are common life-threatening concerns for patients undergoing hemodialysis. Volume overload can induce hypertension, left ventricular hypertrophy, and congestive heart failure and can even lead to mortality [1, 2]. Regarding malnutrition, there are several phenotypes, such as protein-energy wasting (PEW) and sarcopenia [3, 4]. PEW, a malnutritional state characterized by the volume reduction of muscle and fat caused by the decreased intake of energy and/or protein and chronic inflammation [5, 6], is highly prevalent and associated with increased risks of all-cause mortality and cardiovascular mortality in hemodialysis patients [7, 8]. Sarcopenia, a condition defined as the loss of skeletal muscle mass and function, is also prevalent and associated with mortality in this population [9–11]. A common element of PEW and sarcopenia is muscle wasting or loss of muscle; therefore, hemodialysis patients can concomitantly have PEW and sarcopenia [3, 4]. We recently reported that computed tomography-measured sarcopenic indices are promising indicators of mortality in hemodialysis patients [12–15]. According to the Asian Working Group for Sarcopenia: 2019 Consensus Update on Sarcopenia Diagnosis and Treatment, muscle function, such as muscle strength, is considered more important than muscle mass for predicting sarcopenia and mortality in the general population [10]. However, muscle function usually deteriorates, and muscle wasting (loss of muscle mass) is now recognized as an important predictor of mortality in the hemodialysis population [16]. The simplified creatinine index (SCI), which is an objective nutritional assessment tool calculated using sex, age, single-pool Kt/V for urea (urea kinetic modeling based on the assumption that urea is located only in one compartment of the body: spKt/Vurea), and serum creatinine level before a hemodialysis session, has been introduced as a marker of muscle mass volume or muscle wasting [17, 18]. The SCI, which helps to predict infection-related, cardiovascular, and all-cause mortality [19, 20], has also been recognized as a new indicator of PEW [21].

Bioelectrical impedance analysis (BIA) has attracted attention as a useful tool to assess the fluid volume and monitor the nutritional status of the hemodialysis population in daily clinical practice [22, 23]. Recently, the BIA-measured ratio of extracellular water (ECW) to intracellular water (ICW), which can simultaneously reflect fluid overload and malnutrition, has emerged as an indicator of PEW in patients receiving hemodialysis [24]. The ECW/ICW ratio can help predict cardiovascular mortality, cardiovascular events, and all-cause mortality in this population [24–26]. Park et al. reported that ECW fluid overload was associated with sarcopenia in the general population [27]. Additionally, ICW may be representative of body cell or skeletal muscle mass [28]; therefore, we hypothesized that the ECW/ICW ratio might be an indicator of muscle wasting. However, the association between the ECW/ICW ratio and muscle wasting in hemodialysis patients remains unclear.

This study aimed to examine the association between the ECW/ICW ratio and the SCI in hemodialysis patients. Moreover, we evaluated whether the combination of the ECW/ICW ratio and SCI could stratify the risks of all-cause mortality and improve the accuracy of predicting mortality in this population.

## Methods

### Study participants

This retrospective study included patients who had steadily received hemodialysis for more than 6 months (three times per week for four hours) and underwent body composition measurements using BIA at our institution from January 2009 to December 2019. Patient data were anonymized before they were accessed; therefore, the requirement for informed consent

was waived by the ethics committee of our institution. This study adhered to the principles of the Declaration of Helsinki, and the study protocol was approved by the Ethics Committee of Matsunami General Hospital (approval no. 523).

## Baseline data collection

The following patient data were obtained from medical records: age, sex, hemodialysis vintage, cause of end-stage kidney disease, alcohol consumption, tobacco use, and history of hypertension, diabetes mellitus, and cardiovascular events. In this study, hypertension was defined as the use of any anti-hypertensive medications and/or systolic blood pressure ≥140 mmHg and/or diastolic blood pressure ≥90 mmHg before a hemodialysis session. Diabetes mellitus was defined as a history of diabetes mellitus and/or treatment with anti-diabetic agents. Cardiovascular disease (CVD) events were defined as a history of angina pectoris, myocardial infarction, congestive heart failure, stroke, or peripheral artery disease. Blood tests were performed with the patient in the supine position before and after the hemodialysis session on a Monday or Tuesday. BIA and chest radiographs were performed on the same day after the hemodialysis session. Body composition data, including the ECW, ICW, and total body water, were measured using a multi-frequency (2.5–350 kHz) body composition analyzer (MLT-550N; SK Medical, Siga, Japan) with the wrist-ankle method. The body mass index (BMI) was calculated as follows: BMI = dry weight (kg)/height squared ($m^2$). The geriatric nutritional risk index (GNRI) was calculated using the serum albumin level and BMI as follows [29–31]: GNRI = 14.89 × serum albumin level (g/dL) + 41.7 × BMI ($kg/m^2$) / 22. When the BMI was greater than 22 $kg/m^2$, the variable BMI ($kg/m^2$)/22 was set to 1. The SCI was calculated using parameters such as age, sex, single-pool Kt/V for urea (spKt/Vurea), and the pre-hemodialysis serum creatinine level as follows [17]: SCI (mg/kg/day) = 16.21 + 1.12 × (1 for males; 0 for females) − 0.06 × age (years) − 0.08 × spKt/Vurea + 0.009 × pre-hemodialysis creatinine level (μmol/L).

## Follow-up and study endpoint

Patients were divided into two groups based on the cut-off values of the ECW/ICW ratio (0.57) and SCI (20.4 mg/kg/day) for maximumly discriminating survival. Thereafter, they were divided into four subgroups with the following cut-off points: G1, ECW/ICW ratio <0.57 and SCI ≥20.4 mg/kg/day; G2, ECW/ICW ratio <0.57 and SCI <20.4 mg/kg/day; G3, ECW/ICW ratio ≥0.57 and SCI ≥20.4 mg/kg/day; and G4, ECW/ICW ratio ≥0.57 and SCI <20.4 mg/kg/day. Patients were followed-up until December 2020. The study endpoint was all-cause mortality.

## Statistical analysis

Normally distributed variables are expressed as mean ± standard deviation, and non-normally distributed variables are expressed as the median and interquartile range. The cut-off values of the ECW/ICW ratio and SCI for maximally predicting mortality were obtained using receiver operating characteristic (ROC) analysis and the Youden index (the maximum sum of sensitivity and specificity minus one). To compare the differences in the baseline characteristics of the patients who were divided into four subgroups based on the cut-off values of the ECW/ICW ratio and SCI, a one-way analysis of variance or the Kruskal–Wallis test was performed for continuous variables, and the chi-squared test was used for categorical variables. A univariate linear regression analysis was performed to examine the correlations between the ECW/ICW and baseline factors. The multivariate linear regression analysis included factors significantly associated with the ECW/ICW ratio in the univariate analysis. The Kaplan–Meier method was

used to estimate the survival rate, and the differences were analyzed using the log-rank test. A univariate Cox proportional hazard regression analysis was performed to estimate hazard ratios and 95% confidence intervals [CIs] for all-cause mortality. Multivariate Cox proportional hazard regression analysis was performed by adjusting for age, sex, and variables that had a p value of <0.1 in the univariate Cox analysis (hemodialysis vintage, history of CVD events, hemoglobin, phosphorus, C-reactive protein (CRP), and GNRI).

To assess whether the predictive accuracy of all-cause mortality could improve when the ECW/ICW ratio and/or SCI were added to the baseline model, the C-index, net reclassification improvement (NRI), and integrated discrimination improvement (IDI) were calculated. As a baseline model, variables, as mentioned above, were included. The C-index was defined as the area under the ROC curve in the logistic regression analysis [32]. The C-index of the enriched model with the addition of the ECW/ICW ratio and/or SCI was compared to that of the baseline model. The NRI was a relative indicator of the number of patients with improved predicted mortality risk, and the IDI was an indicator of the average improvement in the predicted mortality risk after adding the ECW/ICW ratio and/or SCI to the baseline model [33].

All statistical analyses were performed using R version 4.04 (R Foundation for Statistical Computing, Vienna, Austria) and SPSS version 24 (IBM Corp., Armonk, NY, USA); P<0.05 was considered to be statistically significant.

## Results

### Baseline characteristics

We included 224 patients receiving hemodialysis in the present study. Their baseline characteristics are summarized in Table 1. The mean age was 63.3±13.9 years, and 66.5% of the patients were men. The median hemodialysis vintage was 0.8 years (0.6–4.7 years); 46.9% of the patients had a history of diabetes mellitus, and 62.9% had a history of CVD events. The hemoglobin, total cholesterol, phosphorus, CRP, GNRI, and cardiothoracic ratio (CTR) values were 10.8±1.3 g/dL, 156±35 mg/dL, 5.1±1.3 mg/dL, 0.17 mg/dL (0.06–0.37 mg/dL), 93.2±6.4, and 49.3±5.0%, respectively. The median ECW/ICW ratio and SCI were 0.56 (0.46–0.69) and 19.9 mg/kg/day (18.1–22.1 mg/kg/day), respectively.

### Relationship between the ECW/ICW ratio and SCI

The ECW/ICW ratio was positively correlated with age, male sex, history of diabetes, CTR, and log CRP and was negatively correlated with the GNRI and SCI. In the multivariate regression analysis, the ECW/ICW ratio was independently associated with age (β = 0.201; P = 0.013), male sex (β = 0.298; P <0.0001), diabetes (β = 0.207; P = 0.0003), CTR (β = 0.134; P = 0.024), GNRI (β = -0.271; P<0.0001), and SCI (β = -0.164; P = 0.042) (Table 2).

### Association of the ECW/ICW ratio and/or the SCI with all-cause mortality

During a median follow-up of 3.5 years (2.0–6.0 years), 77 patients died because of the following reasons: CVD (N = 38; 49.4%), infection (N = 22; 28.6%), cancer (N = 10; 13.0%), and other causes (N = 7; 9.1%). The univariate Cox proportional hazard regression analysis revealed that the ECW/ICW ratio (continuous) and the SCI (continuous) were significant predictors of all-cause mortality (ECW/ICW ratio: HR 1.05, 95% CI 1.04–1.06, p <0.0001; SCI: HR 0.83, 95% CI 0.76–0.90, p <0.0001). However, to maximize the predictive value of all-cause mortality, the cut-off points were determined using ROC analysis; ECW/ICW ratio: cut-off value 0.57, AUC 0.737, sensitivity 0.687, specificity 0.741, p <0.0001; SCI: cut-off value 20.4 mg/kg/day, AUC 0.623, sensitivity 0.496, specificity 0.715, p = 0.0006. The 10-year all-cause survival rates were

**Table 1. Baseline patient characteristics.**

| | All patients (*N* = 224) | G1 (*N* = 63) | G2 (*N* = 58) | G3 (*N* = 31) | G4 (*N* = 72) | P value |
|---|---|---|---|---|---|---|
| Age (years) | 63.3 ± 13.9 | 50.2 ± 13.5 | 67.2 ± 10.5 | 64.2 ± 10.5 | 71.4 ± 8.9 | <0.0001 |
| Male sex | 66.5 | 68.3 | 44.8 | 96.7 | 69.4 | <0.0001 |
| Underlying kidney disease | | | | | | 0.033 |
| Diabetic kidney disease | 44.2 | 31.7 | 44.8 | 38.7 | 56.9 | |
| Chronic glomerulonephritis | 29.5 | 42.9 | 31.0 | 32.2 | 15.3 | |
| Nephrosclerosis | 18.8 | 14.3 | 17.2 | 25.8 | 20.8 | |
| Others | 7.6 | 11.1 | 6.9 | 3.2 | 6.9 | |
| HD duration (years) | 0.8 (0.6–4.7) | 1.5 (0.6–9.3) | 0.6 (0.5–1.9) | 5.1 (1.2–9.2) | 0.6 (0.5–1.2) | 0.0002 |
| Alcohol | 21.0 | 22.2 | 20.7 | 25.8 | 18.1 | 0.84 |
| Smoking | 26.8 | 31.7 | 27.6 | 19.4 | 25.0 | 0.61 |
| Hypertension | 95.5 | 93.7 | 94.8 | 93.5 | 98.6 | 0.40 |
| Diabetes mellitus | 46.9 | 31.7 | 51.7 | 45.2 | 56.9 | 0.024 |
| History of CVD | 62.9 | 44.4 | 69.0 | 64.5 | 73.6 | 0.0037 |
| BMI (kg/m$^2$) | 22.0 ± 4.0 | 22.0 ± 3.8 | 22.7 ± 5.1 | 22.1 ± 3.5 | 21.5 ± 3.3 | 0.37 |
| BUN (mg/dL) | 60.5 ± 15.0 | 69.1 ± 17.6 | 58.7 ± 10.5 | 64.1 ± 14.2 | 53.0 ± 11.5 | <0.0001 |
| Creatinine (mg/dL) | 8.9 ± 2.9 | 11.9 ± 2.1 | 7.2 ± 1.8 | 11.1 ± 1.4 | 6.9 ± 1.7 | <0.0001 |
| Albumin (g/dL) | 3.6 ± 0.3 | 3.8 ± 0.3 | 3.7 ± 0.3 | 3.6 ± 0.3 | 3.5 ± 0.4 | <0.0001 |
| Hemoglobin (g/dL) | 10.8 ± 1.3 | 11.1 ± 1.2 | 10.9 ± 1.3 | 10.6 ± 1.1 | 10.5 ± 1.5 | 0.050 |
| T-Cho (mg/dL) | 156 ± 35 | 152 ± 33 | 172 ± 34 | 152 ± 34 | 147 ± 36 | 0.0007 |
| Phosphorus (mg/dL) | 5.1 ± 1.3 | 5.8 ± 1.4 | 4.9 ± 1.1 | 5.3 ± 1.4 | 4.6 ± 1.2 | <0.0001 |
| Glucose (mg/dL) | 138 ± 57 | 132 ± 59 | 139 ± 59 | 136 ± 45 | 143 ± 58 | 0.72 |
| CRP (mg/dL) | 0.17 (0.06–0.37) | 0.11 (0.05–0.21) | 0.18 (0.06–0.32) | 0.23 (0.06–0.55) | 0.21 (0.06–0.53) | 0.034 |
| GNRI | 93.2 ± 6.4 | 95.8 ± 5.6 | 93.6 ± 6.6 | 92.3 ± 5.3 | 90.9 ± 6.3 | <0.0001 |
| Kt/V | 1.3 ± 0.3 | 1.4 ± 0.3 | 1.4 ± 0.3 | 1.3 ± 0.2 | 1.2 ± 0.3 | 0.0013 |
| SCI (mg/kg/day) | 20.1 ± 2.9 | 23.2 ± 2.1 | 18.2 ± 1.5 | 22.0 ± 1.4 | 18.0 ± 1.7 | <0.0001 |
| CTR | 49.3 ± 5.0 | 47.0 ± 4.5 | 50.2 ± 5.3 | 49.0 ± 4.3 | 50.7 ± 5.0 | 0.0001 |
| TBW (kg) | 27.5 ± 5.4 | 28.4 ± 5.8 | 25.1 ± 5.5 | 30.0 ± 4.5 | 27.5 ± 4.5 | 0.0001 |
| ICW (kg) | 17.4 ± 3.7 | 19.8 ± 3.8 | 17.0 ± 3.6 | 17.7 ± 3.0 | 15.5 ± 2.7 | <0.0001 |
| ECW (kg) | 10.0 ± 3.1 | 8.6 ± 2.6 | 8.1 ± 2.2 | 12.3 ± 1.9 | 12.0 ± 2.7 | <0.0001 |
| ECW/ICW ratio | 0.59 ± 0.20 | 0.43 ± 0.10 | 0.47 ± 0.07 | 0.70 ± 0.10 | 0.78 ± 0.19 | <0.0001 |

Data are provided as mean ± standard deviation or median and interquartile range or percentages. A one-way analysis of variance or the Kruskal–Wallis test was performed for continuous variables, and the chi-squared test was performed for categorical variables.

BMI, body mass index; CRP, C-reactive protein; CTR, cardiothoracic ratio; CVD, cardiovascular disease; ECW, extracellular water; GNRI, geriatric nutritional risk index; HD, hemodialysis; ICW, intracellular water; SCI, simplified creatinine index; TBW, total body water.

26.9% in the higher ECW/ICW ratio group and 61.7% in the lower ECW/ICW ratio group (P<0.0001) (Fig 1a); these rates were 26.7% in the lower SCI group and 64.8% in the higher SCI group (P = 0.0002) (Fig 1b). The 10-year all-cause survival rates were 76.6% in group 1 (G1), 43.9% in group 2 (G2), 39.4% in group 3 (G3), and 0% in group 4 (G4) (P<0.0001). After adjusting for age and sex, hemodialysis vintage, history of CVD, hemoglobin, phosphorus, CRP, and GNRI, a higher ECW/ICW ratio (adjusted hazard ratio [aHR], 3.66; 95% CI, 1.99–6.72; P<0.0001) and lower SCI (aHR, 2.25; 95% CI, 1.34–3.79; P = 0.0021) were independently associated with increased risks of all-cause mortality (Table 3). Moreover, compared with G1, the aHRs for G2, G3, and G4 were 4.29 (95% CI, 1.30–14.18; P = 0.017), 8.61 (95% CI, 2.61–28.46; P = 0.0004), and 12.22 (95% CI, 3.68–40.57; P<0.0001), respectively (Table 3).

**Table 2. Regression analysis of the relationships between the ECW/ICW ratio and baseline variables.**

| Variables | Univariate | | Multivariate | |
|---|---|---|---|---|
| | r | P value | β | P value |
| Age | 0.487 | <0.0001 | 0.201 | 0.013 |
| Male sex | 0.216 | 0.0011 | 0.298 | <0.0001 |
| Diabetes | 0.216 | 0.0011 | 0.207 | 0.0003 |
| CTR | 0.267 | <0.0001 | 0.134 | 0.024 |
| Log CRP | 0.207 | 0.0019 | 0.075 | 0.169 |
| GNRI | -0.349 | <0.0001 | -0.271 | <0.0001 |
| SCI | -0.361 | <0.0001 | -0.164 | 0.042 |

The univariate linear regression analysis was performed to examine the correlations between the ECW/ICW and baseline factors. The multivariate linear regression analysis was performed with significant variables in the univariate analysis.

CTR, cardiothoracic ratio; CRP, C-reactive protein; GNRI, geriatric nutritional risk index; SCI, simplified creatinine index.

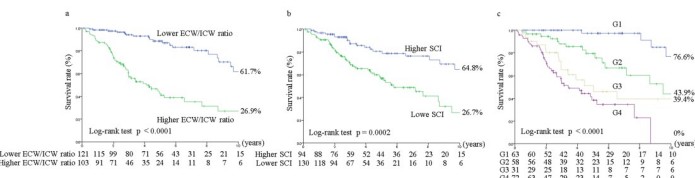

**Fig 1. Kaplan–Meier survival curves for all-cause mortality.** The Kaplan–Meier method was used to estimate the survival rate, and the differences were analyzed using the log-rank test. All-cause survival rates for the two groups of (ratio of extracellular water to intracellular water) the ECW/ICW ratio <0.57 vs. ECW/ICW ratio ≥0.57 (a), two groups of the simplified creatinine index (SCI) <20.4 mg/kg/day vs. mCI ≥20.4 mg/kg/day (b), and four groups of the combined ECW/ICW and SCI (c). G1 (group 1), ECW/ICW ratio <0.57 and SCI ≥20.4 mg/kg/day; G2 (group 2), ECW/ICW ratio <0.57 and SCI <20.4 mg/kg/day; G3 (group 3), ECW/ICW ratio ≥0.57 and SCI ≥20.4 mg/kg/day; and G4 (group 4), ECW/ICW ratio ≥0.57 and SCI <20.4 mg/kg/day.

**Table 3. Cox proportional hazards analysis of the ECW/ICW ratio and the SCI for mortality.**

| Variables | Univariate | | Multivariate* | |
|---|---|---|---|---|
| | HR (95% CI) | P value | HR (95% CI) | P value |
| Lower SCI (SCI <20.4 mg/kg/day) | 2.73 (1.64–4.53) | 0.0001 | 2.25 (1.34–3.79) | 0.0021 |
| Higher ECW/ICW ratio (ECW/ICW ratio ≥0.57) | 5.33 (3.17–8.95) | <0.0001 | 3.66 (1.99–6.72) | <0.0001 |
| Cross-classified (vs. G1) | | <0.0001 | | <0.0001 |
| G2 | 5.69 (1.89–17.10) | 0.0019 | 4.29 (1.30–14.18) | 0.017 |
| G3 | 11.47 (3.87–33.99) | <0.0001 | 8.61 (2.61–28.46) | 0.0004 |
| G4 | 20.48 (7.11–58.95) | <0.0001 | 12.22 (3.68–40.57) | <0.0001 |

The univariate Cox proportional hazard regression analysis was performed to estimate hazard ratios and 95% CIs for all-cause mortality. Multivariate Cox proportional hazard regression analysis was performed by adjusting for age, sex, and variables that had a p value of <0.1 in the univariate Cox analysis (hemodialysis vintage, history of CVD, hemoglobin, phosphorus, C-reactive protein, and geriatric nutritional risk index).

HR, hazard ratio; CI, confidence interval; SCI, simplified creatinine index; ECW, extracellular water; ICW, intracellular water.

## Model discrimination

The addition of the SCI alone and ECW/ICW ratio alone into the baseline risk model, including age, sex, hemodialysis vintage, history of CVD, hemoglobin, phosphorus, CRP, and GNRI,

**Table 4. Predictive accuracy of the ECW/ICW ratio and the SCI for all-cause mortality.**

| Variables | C-index | P value | NRI | P value | IDI | P value |
|---|---|---|---|---|---|---|
| Established risk factors* | 0.831 (0.777–0.886) | Ref. | Ref. | | Ref. | |
| + SCI | 0.838 (0.785–0.891) | 0.31 | 0.348 | 0.011 | 0.008 | 0.24 |
| + ECW/ICW ratio | 0.858 (0.808–0.907) | 0.086 | 0.893 | <0.0001 | 0.056 | 0.0006 |
| + SCI and ECW/ICW ratio | 0.864 (0.816–0.912) | 0.045 | 0.882 | <0.0001 | 0.062 | 0.0002 |

The C-index, net reclassification improvement (NRI), and integrated discrimination improvement (IDI) were calculated to investigate the predictive accuracy of all-cause mortality. Established risk factors* included age, sex, hemodialysis vintage, history of cardiovascular disease, hemoglobin, phosphorus, C-reactive protein, and geriatric nutritional risk index.

NRI, net reclassification improvement; IDI, integrated discrimination improvement; SCI, simplified creatinine index; ECW, extracellular water; ICW, intracellular water.

did not improve the C-index for predicting all-cause mortality; however, the addition of both significantly improved the C-index from 0.831 to 0.864 (P = 0.045) (Table 4).

## Discussion

This study demonstrated that the ECW/ICW ratio was independently associated with the SCI and that a higher ECW/ICW ratio and lower SCI were independently associated with increased risks of all-cause mortality. Moreover, the combination of the ECW/ICW ratio and SCI significantly improved the predictability of all-cause mortality and stratified the risk of mortality. These findings suggest that the ECW/ICW ratio may be an indicator of muscle wasting and PEW and support the importance of assessing both the ECW/ICW ratio and SCI for predicting mortality in the hemodialysis population.

In this study, the ECW/ICW ratio was independently associated with diabetes, CTR, GNRI, and SCI. Nakajima et al. reported that the ECW/ICW ratio was independently associated with albuminuria levels in patients with type 2 diabetes mellitus without renal failure and suggested that the ECW/ICW ratio could be helpful to monitor fluid volume imbalance in patients with type 2 diabetes [34]. Additionally, we previously reported the independent association of the ECW/ICW ratio with CTR and GNRI, reflecting volume expansion and malnutrition, respectively [24]. GNRI, which helps to assess the longitudinal nutritional status and predict mortality, is an objective marker of PEW in patients undergoing hemodialysis and is easy to calculate [35]. Moreover, in this study, the ECW/ICW ratio was negatively and independently associated with the SCI. The SCI has been developed as a surrogate marker of muscle mass in hemodialysis patients with anuria [17]. Tsai et al. recently reported that SCI was an independent predictor of PEW [21]. Yamada et al. reported that SCI was highly associated with muscle mass volume, measured using BIA and anthropometry [36]. Furthermore, Yamamoto et al. reported that the SCI was correlated with handgrip strength and gait speed [37]. Therefore, in this study, the independent association of the ECW/ICW ratio with GNRI and SCI suggests that the ECW/ICW ratio may be an indicator of not only PEW but also sarcopenia. However, to clarify the relationship between the ECW/ICW ratio and sarcopenia, direct associations of the ECW/ICW ratio with skeletal muscle mass volume, muscle strength, and gait speed must be investigated in the future. Since the ICW may be used to estimate the skeletal muscle mass volume, methods other than BIA, such as dual-energy X-ray absorptiometry or computed tomography, may be recommended as reference methods.

Possible mechanisms that may explain the association between the ECW/ICW ratio and muscle wasting or sarcopenia have been considered. Fluid overload, which is an increase in

the ECW, can lead to intestinal edema and may induce inflammation via the translocation of bowel endotoxin into the circulation [38]. This inflammatory process leads to malnutrition caused by protein catabolism and muscle wasting [39, 40]. Moreover, the fluid overload may impair the absorption of bowel nutrients, such as protein and vitamin D, secondary to bowel edema and may lead to decreased muscle mass and function [41, 42]. Furthermore, a decrease in ICW itself may directly reflect a decrease in muscle mass [28].

In this study, patients with a higher ECW/ICW ratio and those with a lower SCI were independently associated with an increased risk of all-cause mortality in patients undergoing hemodialysis. The ROC-derived cut-off values of the ECW/ICW ratio and SCI for maximally predicting all-cause mortality were 0.57 and 20.4 mg/kg/day, respectively. Kim et al. reported that the cut-off value for maximum discrimination of survival was 0.57 in hemodialysis patients. Canaud et al. demonstrated that the SCI was higher in men than in women, and a higher SCI ($>$20.5 mg/kg/day) not categorized by sex was independently associated with a decreased risk of mortality in the hemodialysis population. Thus, the cut-off values in the present study were consistent with those that were previously reported. However, the proportion of men in G3 (the group with a higher ECW/ICW ratio and a higher SCI) was relatively high. Therefore, we included sex as a covariate in the multivariate Cox proportional hazard regression analysis, similar to Canaud et al.'s study. Furthermore, patients with a higher ECW/ICW ratio and a lower SCI were at the highest risk of all-cause mortality in this population. Regarding model discrimination, compared to the baseline risk model, including the GNRI, the addition of both the ECW/ICW ratio and the SCI significantly improved the C-index. Therefore, this study demonstrated that the combined evaluation of the ECW/ICW ratio and SCI enabled the stratification of the risk of all-cause death and improved mortality prediction.

There were some limitations to this study. First, this single-center retrospective study included a relatively small number of patients receiving hemodialysis. In a previous study, the sample size that was used to investigate the relationship of the ECW/ICW ratio or the ECW/ total body water (TBW) ratio with mortality in dialysis patients had a median of 152 (77–234) [minimum: 53 (peritoneal dialysis); maximum: 753 (388 hemodialysis and 365 peritoneal dialysis)] (N = 11) [43]. Moreover, the median hemodialysis vintage was less than one year. Therefore, our findings may not be applicable to patients with a longer hemodialysis duration. Second, the number of events (death) was small; therefore, full adjustments with covariables were difficult in the multivariate Cox analysis. Third, the ECW/ICW ratio and the SCI were measured only at the time of study enrollment; therefore, changes in these values during follow-up periods were not considered. Fourth, this study included only Japanese patients receiving hemodialysis; therefore, the findings of this study may not be applicable to all patients receiving hemodialysis in other countries. Fifth, residual kidney function, which may affect the SCI value, could not be evaluated because of the retrospective nature of this study. Further prospective, large-scale, multicenter studies may be required to validate our study findings.

## Conclusions

The ECW/ICW ratio was independently associated with the SCI, and both the ECW/ICW ratio and the SCI independently predicted all-cause mortality for patients receiving hemodialysis. Moreover, the combination of the ECW/ICW ratio and SCI was useful for stratifying the mortality risk and improving the accuracy of the mortality prediction. Therefore, the ECW/ICW ratio may be an indicator of muscle wasting, and the combination of the ECW/ICW ratio and SCI may help accurately predict all-cause mortality in this population.

## Supporting information

**S1 File.**
(XLSX)

## Author Contributions

**Conceptualization:** Takahiro Yajima, Kumiko Yajima.

**Investigation:** Takahiro Yajima, Kumiko Yajima.

**Methodology:** Takahiro Yajima.

**Supervision:** Kumiko Yajima.

**Validation:** Takahiro Yajima, Kumiko Yajima.

**Visualization:** Takahiro Yajima.

**Writing – original draft:** Takahiro Yajima.

**Writing – review & editing:** Takahiro Yajima, Kumiko Yajima.

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
