## [Decision Letter · Decision Letter 0]

8 Feb 2023

PONE-D-23-00211Ratio of extracellular water to intracellular water and simplified creatinine index as predictors of all-cause mortality for patients receiving hemodialysisPLOS ONE

Dear Dr. Yajima,

Thank you for submitting your manuscript to PLOS ONE. After careful consideration, we feel that it has merit but does not fully meet PLOS ONE’s publication criteria as it currently stands. Therefore, we invite you to submit a revised version of the manuscript that addresses the points raised during the review process.

We look forward to receiving your revised manuscript.

Kind regards,

William M. Adams

Academic Editor

PLOS ONE

Journal Requirements:

e) Please provide an amended Funding Statement that declares *all* the funding or sources of support received during this specific study (whether external or internal to your organization) as detailed online in our guide for authors at http://journals.plos.org/plosone/s/submit-now.  

f) Please state what role the funders took in the study.  If any authors received a salary from any of your funders, please state which authors and which funder. If the funders had no role, please state: "The funders had no role in study design, data collection and analysis, decision to publish, or preparation of the manuscript." 

Please send your amended statements by return email; we will change the online submission form on your behalf. 

Reviewers' comments:

Reviewer's Responses to Questions

**Comments to the Author**

1. Is the manuscript technically sound, and do the data support the conclusions?

Reviewer #1: Yes

Reviewer #2: Yes

2. Has the statistical analysis been performed appropriately and rigorously? 

Reviewer #1: Yes

Reviewer #2: Yes

3. Have the authors made all data underlying the findings in their manuscript fully available?

Reviewer #1: Yes

Reviewer #2: Yes

4. Is the manuscript presented in an intelligible fashion and written in standard English?

Reviewer #1: Yes

Reviewer #2: Yes

5. Review Comments to the Author

Reviewer #1: The manuscript entitled "Ratio of extracellular water to intracellular water and simplified creatinine index as predictors of all-cause mortality for patients receiving hemodialysis" discusses a trending topic that is the value of BIA-assessed fluid compartments ratio to predict all-cause mortality in different clinical conditions. Despite the sample size is limited compared to other studies and the retrospective nature of the study, these limitations have been highlighted within the discussion and the scientific quality of the research worth consideration.

I have some comments/suggestions:

- Abstract: line 43 - maker -> marker

- Introduction:

i) in some parts of the text the authors refer to muscle function that, unfortunately, was not measured within this study. Can the authors better describe the association between muscle function and muscle mass (and in case, ECW/ICW), otherwise it seems to me poorly linked and useful in consideration of the actual manuscript.

ii) since the readership of this journal can be various, I recommend to first define the single-pool Kt/V for urea as single-pool kinetic modeling

- Methods: although the limitation has been already presented in the discussion, can the authors provide any justification for the sample size, or power analysis? It could strengthen the quality of this work.

- Results: for better clarity, I suggest to add in the notes of the tables and figures what statistical test was used to assess the significance

Discussion:

i) line 256 - "more specific surrogate marker", more specific compared to what? I think that specificity should be assessed in detail as it has a clear statistical meaning.

ii) the authors considered BMI in their measures and models; some research has highlighted how ECW/ICW could be associated with body fat and obesity, and the latter can predict mortality; can the authors comment on that? From their data it seems that ECW/ICW was not associated with BMI, and BMI was not a predictor of mortality; is it correct?

iii) the analyses were performed based on the median values from this sample. Do the authors know if such values (e.g., 0.559 ECW/ICW) can be common in the general population or other diseases?

Reviewer #2: General comments

The present retrospective study examined the differential impact of ECW/ICW ratio and simplified creatinine (Cr) index (SCI) on all-cause mortality in 224 Japanese patients undergoing hemodialysis. The authors used EWC/ICW as a marker for volume overload and/or sarcopenia and SCI as a proxy for skeletal muscle mass. The authors showed that both ECW/ICW ratio and SCI were independently associated with all-cause death, by using multi-variable adjusted Cox proportional hazard regression analyses. Furthermore, they showed that the combination of ECW/ICW and SCI improved predictability of future death by calculating C-statistics, IDI, and NRI.

The paper is simple, concise, and well-written. The statistical analyses appear to be correct and appropriately applied. I have some comments on this paper.

Specific comments

#. It is important to show some cut-off values of ECW/ICW to know the risk of death. The association between 1 unit change in each marker and risk of death is less useful for clinicians.

#. Table 3. Readers may misunderstand that all the parameters listed in Table 3 were included in the same model. Some of them are inter-correlated. I propose the data presentation should be changed more appropriately.

#. The reasons for the selection of covariates in the multivariable model should be described in the Methods section. My question is why some were selected and others were not.

#. The number of covariates included in the multivariable model appears to be too many for the outcome number. The outcome number was 77. The variables used in the present study were from 10 to 12. The variable of cross-classified one (G1-G4) included three variables. If the authors wanted to use many covariates in the same model, potential limitations should be described in the limitation section.

#. SCI is dependent of skeletal muscle mass. SCI is closely associated with serum creatinine value and serum creatinine value reflects skeletal muscle mass in anuric patients. Hence, the mean and median values of SCI in men are expected to be higher than those in women. In this regard, the cut-off value of 19.87 mg/kg/day may not be appropriate. The median value would be different across men and women in the present study. It would be better to use sex-specific median values for categorization. Actually, female patients were more in G2 and G4. This point should be improved

#. SCI is mainly determined by serum Cr level. The calculation of SCI is cumbersome. Why not use serum creatinine level instead of SCI? SCI may be better to use in the clinical research because correlation coefficient is generally higher in SCI than serum Cr levels. However, serum Cr levels are simpler for medical practitioners. Combination of ECW/ICW and serum Cr level may provide almost the same results in the present study. It would be interesting to show if SCI could be replaced with serum Cr or not. Serum Cr may be used as a continuous variable or nominal variable.

#. The median value of dialysis duration was less than one year. The present data may not be applicable to those with a longer dialysis duration. Namely, the generalizability of the paper may be limited. The present hemodialysis population may not be representative common hemodialysis patients. This point should be described in the limitation section.

#. The dialysis time per session should be included in Table 1.

6. PLOS authors have the option to publish the peer review history of their article (what does this mean?). If published, this will include your full peer review and any attached files.

Reviewer #1: No

Reviewer #2: No

---

## [Author Response · Author response to Decision Letter 0]

17 Feb 2023

Response to Reviewer #1

Thank you very much for your constructive comments.

Reviewer #1: The manuscript entitled "Ratio of extracellular water to intracellular water and simplified creatinine index as predictors of all-cause mortality for patients receiving hemodialysis" discusses a trending topic that is the value of BIA-assessed fluid compartments ratio to predict all-cause mortality in different clinical conditions. Despite the sample size is limited compared to other studies and the retrospective nature of the study, these limitations have been highlighted within the discussion and the scientific quality of the research worth consideration.

I have some comments/suggestions:

- Abstract: line 43 - maker -> marker

Thank you for your valuable comment. According to the reviewer's suggestions, we have revised the manuscript accordingly.

- Introduction:

i) in some parts of the text the authors refer to muscle function that, unfortunately, was not measured within this study. Can the authors better describe the association between muscle function and muscle mass (and in case, ECW/ICW), otherwise it seems to me poorly linked and useful in consideration of the actual manuscript.

Thank you for this helpful suggestion. In the revised manuscript, we have described the association between muscle mass and muscle function in the general population and the hemodialysis population, respectively. Because ECW overload may be associated with sarcopenia, and ICW can reflect muscle mass, we hypothesized that the ECW/ICW ratio might be an indicator of loss of muscle mass. Accordingly, we have added this information in the Introduction section.

ii) since the readership of this journal can be various, I recommend to first define the single-pool Kt/V for urea as single-pool kinetic modeling

Thank you for this valuable comment. In the revised manuscript, we have added the definition of single-pool Kt/V for urea.

- Methods: although the limitation has been already presented in the discussion, can the authors provide any justification for the sample size, or power analysis? It could strengthen the quality of this work.

Thank you very much for your valuable comments.

We calculated the sample size using EZR, and the estimated number of patients needed was 140. However, we understand that calculation of the sample size is done only to plan prospective studies; since our study had a retrospective nature, calculating the sample size was not essential. Thus, we reviewed previous studies to investigate the relationship of the ECW/ICW ratio or ECW/TBW ratio, another marker of body composition, with mortality in dialysis (PD and HD) patients, and we found that the sample size used was a median of 152 (IQR 77-234) [minimum: 53 (PD); maximum: 753 (388 HD and 365 PD)] (N=11). 

- Results: for better clarity, I suggest to add in the notes of the tables and figures what statistical test was used to assess the significance

Thank you for your valuable suggestion. In accordance with your suggestion, the statistical methods in the notes of the tables and figures have been revised.

Discussion:

i) line 256 - "more specific surrogate marker", more specific compared to what? I think that specificity should be assessed in detail as it has a clear statistical meaning.

Thank you very much for the comments.

We agree with your comments. We sincerely apologize that our original paper did not clearly communicate our intended meaning.

We have added the following sentence in the discussion section (Page 20, lines 294-296). "Therefore, in this study, the independent association of the ECW/ICW ratio with GNRI and SCI suggests that the ECW/ICW ratio may be an indicator of not only PEW but also sarcopenia." 

Thank you for your kind understanding.

ii) the authors considered BMI in their measures and models; some research has highlighted how ECW/ICW could be associated with body fat and obesity, and the latter can predict mortality; can the authors comment on that? From their data it seems that ECW/ICW was not associated with BMI, and BMI was not a predictor of mortality; is it correct?

Thank you for this question. The following points should be considered to answer this question.

In the present study, the ECW/ICW ratio was independently associated with GNRI, a combined predictor of serum albumin and BMI. In the univariate regression analysis, the ECW/ICW was not correlated with BMI (r=-0.038, p=0.57), whereas the ECW/ICW was significantly correlated with body fat percentage (r=-0.150, p=0.025). In multivariate regression analysis, the ECW/ICW ratio was independently associated with body fat percentage (β=-0.140, p=0.039). However, the body composition analyzer, which was used in this study, uses ECW and ICW for estimating fat and muscle. Thus, another modality, such as DEXA or CT, must be used to evaluate fat and muscle mass to investigate the associations with the ECW/ICW ratio. Therefore, we believe that a completely different study is needed to investigate the association between ECW/ICW ratio and body fat and/or obesity in the future. 

In this study, an increased BMI tended to be a decreased risk of all-cause mortality (HR 0.94 95%CI 0.88-1.00). This is not contradictory to the obesity paradox. 

iii) the analyses were performed based on the median values from this sample. Do the authors know if such values (e.g., 0.559 ECW/ICW) can be common in the general population or other diseases?

Thank you for this question 

In the general population, ICW: ECW = 3:2. Therefore, the ECW/ICW ratio is 0.667. In the dialysis population, Kim et al. reported that the ECW/ICW ratio was 0.56±0.06, and the ROC-derived cutoff value for maximum discrimination of survival was 0.57. In the present study, the median of the ECW/ICW ratio was 0.56. In addition, the ROC-derived cutoff value of the ECW/ICW ratio was 0.57 (AUC 0.737, sensitivity 0.687, specificity 0.741, p<0.0001). Thus, the cutoff value of the ECW/ICW ratio in this study is consistent with the previous study. Reviewer #2 pointed out the importance of showing the cutoff point of ECW/ICW ratio; therefore, we used the cutoff point of 0.57 in the present study. Thus, we have revised our manuscript thoroughly.

We would like to thank the reviewer for the thoughtful suggestions and insights to further improve the quality of the manuscript.

Response to Reviewer #2

Thank you very much for your constructive comments.

Reviewer #2: General comments

The present retrospective study examined the differential impact of the ECW/ICW ratio and simplified creatinine (Cr) index (SCI) on all-cause mortality in 224 Japanese patients undergoing hemodialysis. The authors used EWC/ICW as a marker for volume overload and/or sarcopenia and SCI as a proxy for skeletal muscle mass. The authors showed that both ECW/ICW ratio and SCI were independently associated with all-cause death by using multi-variable adjusted Cox proportional hazard regression analyses. Furthermore, they showed that the combination of ECW/ICW and SCI improved the predictability of future deaths by calculating C-statistics, IDI, and NRI.

The paper is simple, concise, and well-written. The statistical analyses appear to be correct and appropriately applied. I have some comments on this paper.

Specific comments

#. It is important to show some cutoff values of ECW/ICW to know the risk of death. The association between 1 unit change in each marker and risk of death is less useful for clinicians.

Thank you very much for the comments. The following points can be considered for further clarification.

The cutoff value of the ECW/ICW ratio derived from ROC analysis was 0.57 (AUC 0.737, sensitivity 0.687, specificity 0.741, p<0.0001). In the univariate Cox analysis, a higher ECW/ICW ratio (the ECW/ICW ratio >0.57) was significantly associated with an increased risk of all-cause mortality (HR 5.33, 95%CI 3.17-8.95, p<0.0001). Thus, similar results were obtained. As for the cutoff points of the ECW/ICW ratio, we have detailed these in the discussion section.

#. Table 3. Readers may misunderstand that all the parameters listed in Table 3 were included in the same model. Some of them are inter-correlated. I propose the data presentation should be changed more appropriately.

Thank you very much for the comments. According to the reviewer's suggestions, for the ease of understanding of the readers, we have only included categorical variables in Table 3.

#. The reasons for the selection of covariates in the multivariable model should be described in the Methods section. My question is why some were selected and others were not.

clinically important 

Thank you for this helpful suggestion. We adjusted for the clinically important variables that were previously reported in the multivariate model. However, as you also mentioned in your following comments, the number of events was small in the present study. Therefore, full adjustments are difficult in the present study, which is a limitation of the study. We have added this in the limitation section. In addition, the selected variables might be arbitrary, as you mentioned. Therefore, in the multivariate Cox analysis, we included age and sex and variables that were p <0.1 in the univariate Cox analysis. We have added this information in the method and discussion section and revised our manuscript.

#. The number of covariates included in the multivariable model appears to be too many for the outcome number. The outcome number was 77. The variables used in the present study were from 10 to 12. The variable of cross-classified one (G1-G4) included three variables. If the authors wanted to use many covariates in the same model, potential limitations should be described in the limitation section.

Thank you very much for the comments.

As we mentioned above, the number of events was small; therefore, full adjustments were difficult in the multivariate Cox analysis. To avoid the arbitrary selection of covariates, we included sex, age, and variables that were p<0.1 in the univariate Cox analysis in the multivariate model. We have added this information in the methods and limitation section. 

#. SCI is dependent of skeletal muscle mass. SCI is closely associated with serum creatinine value and serum creatinine value reflects skeletal muscle mass in anuric patients. Hence, the mean and median values of SCI in men are expected to be higher than those in women. In this regard, the cutoff value of 19.87 mg/kg/day may not be appropriate. The median value would be different across men and women in the present study. It would be better to use sex-specific median values for categorization. Actually, female patients were more in G2 and G4. This point should be improved

We sincerely appreciate the reviewer for this helpful suggestion. 

The values of SCI for women and men were 18.83 (16.93-20.74) and 20.39 (18.63-22.45), respectively. As you mentioned, the median values of SCI were significantly higher in men than in women (p <0.0001). When the SCI was divided by the median in each sex, the lower SCI (women: <18.83 mg/kg/day; men: <20.39 mg/kg/day) was significantly associated with an increased risk of all-cause mortality (HR 2.73, 95%CI 1.68-4.42).

However, this is not surprising; Professor Canaud et al., who proposed the SCI, previously reported that "the mean and median values of SCI in men are expected to be higher than those in women." They proposed the cutoff point of the SCI in the overall dialysis population; a higher SCI (>20.5 mg/kg/day) was independently associated with a decreased mortality risk. Because the SCI is a combined predictor of sex, age, serum creatinine, and Kt/V for urea, the SCI is already adjusted by sex. Therefore, there is no need for division by sex, and we believe that this is the merit of the SCI. Moreover, you suggested the importance of cutoff points of variables for predicting mortality; therefore, we determined the cutoff points of the SCI for predicting mortality using ROC analysis. Interestingly, the cutoff value was 20.4 mg/kg/day, which was almost the same as that proposed by Professor Canaud et al. Therefore, we used the cutoff point in this study. However, as you mentioned, the proportion of men in G3 was quite high; therefore, we included sex as a covariate in the multivariate Cox proportional hazard regression analysis, similar to Professor Canaud et al.’s study.

We have revised our manuscript thoroughly and commented on the cutoff value of the SCI in the discussion section. Thank you for your kind understanding of this.

#. SCI is mainly determined by serum Cr level. The calculation of SCI is cumbersome. Why not use serum creatinine level instead of SCI? SCI may be better to use in the clinical research because correlation coefficient is generally higher in SCI than serum Cr levels. However, serum Cr levels are simpler for medical practitioners. Combination of ECW/ICW and serum Cr level may provide almost the same results in the present study. It would be interesting to show if SCI could be replaced with serum Cr or not. Serum Cr may be used as a continuous variable or nominal variable.

Thank you very much for the comments.

Serum creatinine levels for women and men were 8.23 (6.73-10.34) mg/dL and 9.13 (7.08-11.30) mg/dL, respectively. In univariate Cox analysis, serum creatinine (continuous) and lower serum creatinine (women: <8.23 mg/dL; men: <9.13 mg/dL) levels were significantly associated with an increased risk of all-cause mortality (HR 0.87, 95%CI 0.80-0.94, p=0.0003; and HR 1.73, 95%CI 1.09-2.74, p=0.019). However, in the multivariate Cox analysis, adjusted for age, sex, and variables that were p <0.1 in the univariate Cox analysis, lower serum creatinine was not an independent predictor of all-cause mortality (adjusted HR 1.38, 95%CI 0.81-2.37, p=0.24). Thus, we believe that SCI cannot be simply replaced by serum creatinine. 

#. The median value of dialysis duration was less than one year. The present data may not be applicable to those with a longer dialysis duration. Namely, the generalizability of the paper may be limited. The present hemodialysis population may not be representative common hemodialysis patients. This point should be described in the limitation section.

Thank you for this helpful suggestion. We agree with your comments. Thus, in the revised manuscript, we have added the following sentence to the limitation. "The median hemodialysis vintage was less than one year. Therefore, our findings may not be applicable to patients with a longer hemodialysis duration."

#. The dialysis time per session should be included in Table 1.

Thank you very much for the comments. In our study, we included maintenance hemodialysis patients who received hemodialysis three times per week for four hours. We have added this information in the methods section.

We would like to thank the reviewer for the thoughtful suggestions and insights to further improve the quality of the manuscript.

---

## [Decision Letter · Decision Letter 1]

27 Feb 2023

Ratio of extracellular water to intracellular water and simplified creatinine index as predictors of all-cause mortality for patients receiving hemodialysis

PONE-D-23-00211R1

Dear Dr. Yajima,

We’re pleased to inform you that your manuscript has been judged scientifically suitable for publication and will be formally accepted for publication once it meets all outstanding technical requirements.

Kind regards,

William M. Adams

Academic Editor

PLOS ONE

Additional Editor Comments (optional):

Reviewers' comments:

Reviewer's Responses to Questions

**Comments to the Author**

1. If the authors have adequately addressed your comments raised in a previous round of review and you feel that this manuscript is now acceptable for publication, you may indicate that here to bypass the “Comments to the Author” section, enter your conflict of interest statement in the “Confidential to Editor” section, and submit your "Accept" recommendation.

Reviewer #1: All comments have been addressed

Reviewer #2: All comments have been addressed

2. Is the manuscript technically sound, and do the data support the conclusions?

Reviewer #1: Yes

Reviewer #2: Yes

3. Has the statistical analysis been performed appropriately and rigorously? 

Reviewer #1: Yes

Reviewer #2: Yes

4. Have the authors made all data underlying the findings in their manuscript fully available?

Reviewer #1: Yes

Reviewer #2: Yes

5. Is the manuscript presented in an intelligible fashion and written in standard English?

Reviewer #1: Yes

Reviewer #2: Yes

6. Review Comments to the Author

Reviewer #1: I would like to thank the reviewers for addressing my comments and giving appropriate explanations that helped to answer my questions.

I am happy with the proposed amendments and again, I would like to thank the authors and editors for the opportunity to review this interesting manuscript.

Reviewer #2: I have read the revised version of the paper. I admire the authors for their great efforts and attitude to each specific comment. The authors have extensively appropriately revised the paper in response to reviewers’ comments. The quality of the paper is now greatly improved and the revised paper has become more readable and can provide medical practitioners managing hemodialysis patients with useful information for the assessment of nutritional status and sarcopenia. I believe that the paper is now ready for publication in the journal.

7. PLOS authors have the option to publish the peer review history of their article (what does this mean?). If published, this will include your full peer review and any attached files.

Reviewer #1: No

Reviewer #2: **Yes: **Shunsuke Yamada

---

## [Editor Report · Acceptance letter]

1 Mar 2023

PONE-D-23-00211R1 

Ratio of extracellular water to intracellular water and simplified creatinine index as predictors of all-cause mortality for patients receiving hemodialysis 

Dear Dr. Yajima:

I'm pleased to inform you that your manuscript has been deemed suitable for publication in PLOS ONE. Congratulations! Your manuscript is now with our production department. 

Kind regards, 

on behalf of

Dr. William M. Adams 

Academic Editor

PLOS ONE